# Clinical Characteristics and In Silico Analysis of Cystinuria Caused by a Novel *SLC3A1* Mutation

**DOI:** 10.3390/genes13112173

**Published:** 2022-11-21

**Authors:** Lexin Liu, Zihao Xu, Yuelin Guan, Ying Zhang, Xue Li, Yunqing Ren, Lidan Hu, Xiang Yan

**Affiliations:** 1Department of Urology, Pediatric Urolith Center, The Children’s Hospital, Zhejiang University School of Medicine, National Clinical Research Center for Child Health, Hangzhou 310052, China; 2The Children’s Hospital, Zhejiang University School of Medicine, National Clinical Research Center for Child Health, Hangzhou 310052, China; 3Institute of Translational Medicine, Zhejiang University School of Medicine, 268 Kaixuan Road, Hangzhou 310029, China; 4Department of Big Data in Health Science School of Public Health and The Second Affiliated Hospital, Zhejiang University School of Medicine, Hangzhou 310058, China; 5Department of Dermatology, The Children’s Hospital, Zhejiang University School of Medicine, National Clinical Research Center for Child Health, Hangzhou 310052, China

**Keywords:** *SLC3A1*, cystinuria, cystine stone, genetic mutation, whole exome sequencing

## Abstract

Cystinuria is a genetically inherited disorder of renal and intestinal transport, featured as a high concentration of cystine in the urine. Cumulative cystine in urine would cause the formation of kidney stones, which further leads to renal colic and dysfunction. Gene screens have found that mutations in *SLC3A1 or SLC7A9* gene are responsible for most cases of cystinuria, for encoding defective cystine transporters. Here, we presented the genotypic and phenotypic characteristics of one unique case of a three-generation Chinese family. The proband developed severe urolithiasis combined with renal damage. The radiography and computed tomography (CT) scan showed calculus in the left pelvic kidney. Postoperative stone analysis revealed that the stones were mainly composed of cystine. Therefore, to explore its pathogenesis, next-generation Whole Exome Sequencing (WES) and Sanger sequencing identify the proband mutated gene of the proband’s family. In this article, we reported novel compound heterozygous mutations (c.818G>A and c.1011G>A) of the *SLC3A1* gene in a 5-year-old child suffering from a cystine stone from a three-generation family. Bioinformatic analysis was used to predict the pathogenicity and conservation of the target mutation. Conservative sequence and evolutionary conservation analysis indicated that cystine^273^ and proline^337^ were highly conserved among species, and both mutations listed here (Cys273Tyr and Pro337Pro) were pathogenic. To conclude, our study expands the phenotypic and genotypic spectrum of *SLC3A1* and indicates that genetic screening should be considered in the clinic to provide more effective and precise treatment for cystinuria.

## 1. Introduction

Urolithiasis refers to a condition in which stones are in the kidneys, ureters, bladder, or urethra. It is a highly prevalent disease with a considerably increasing incidence worldwide [1]. The symptoms of urolithiasis vary from patient to patient, depending on the stone’s position, size, and shape. The most common symptoms are flank pain, hematuria, nausea, and vomiting [2]. If not adequately managed, patients are prone to renal failure. It is important to note that patients with kidney stones have lower renal function compared to other stone formers [3]. Among urinary stones, cystine stones have the highest recurrence rate (83% within 5 years), which predisposes patients to renal injury due to infection or mechanical damage [4]. Accordingly, the continuous stimulation of cystine stones results in progressive deterioration of renal function, indicating that early diagnosis of cystine stones is of great importance [5]. The incidence of cystine stones in children is 6–10%, which is several times larger than in adults (1%) [6]. Taken together, cystine stones can cause public health problems and economic burdens to individuals, families, and society. It is noteworthy that there is currently no effective treatment to cure cystinuria. Medications for clinical events are mostly for relieving symptoms, such as fluid intake increase, urinary alkalization, and surgery [5,7]. Although clinical management, to some extent, decreases the occurrence of stones, the compliance of cystine stone patients is poor due to the side effects of long-term use [8]. Indeed, numerous cystine stone patients experience repeated surgeries after the initial surgery [9]. Therefore, in-depth insights into the mechanism of cystine stones are required for a fundamental treatment.

Cystinuria is a disorder featured by high levels of cystine in the urine. The dysregulation of a heterotetrameric transporter in renal proximal tubules decreases the uptake of dibasic amino acids (cystine, lysine, arginine, and ornithine), leading to their accumulation in urine [10]. Given the low solubility of cystine in urine under normal pH, high levels of cystine contribute to the formation of cystine stones [11,12].

Cystinuria is an autosomal recessive hereditary disorder, and it is the most common genetic disease in urolithiasis [13]. Two genes have been identified to be involved in cystinuria formation, i.e., *SLC3A1* and *SLC7A9*. *SLC3A1* is located on chromosome 2 and encodes a 685 amino acid glycoprotein rBAT, while the *SLC7A9* is located on chromosome 19 and encodes a 487 amino acid glycoprotein b^0, +^AT [14]. The heavy subunit rBAT is responsible for trafficking, while the light subunit b^0, +^AT is vital for catalyzation [14]. Cystinuria can be categorized into three types: *SLC3A1* mutation is known as type A; *SLC7A9* mutation is known as type B; both *SLC3A1* and *SLC7A9* mutations are known as type AB [15]. To date, a total of 210 clinical mutations of *SLC3A1* have been documented by the Human Gene Mutation Database (HGMD), and genetic diversity of the population has been reported in different ethnic groups [16,17,18], indicating substantial heterogeneity in the genetic basis of cystinuria. A landscape of the genetic mutations related to cystinuria would contribute to the progression of gene therapy for abnormal cystine sedimentation.

In this article, we reported a 5-year-old child diagnosed with a cystine stone from a three-generation Chinese family. A novel compound heterozygous mutation (c.818G>A and c.1011G>A) of the *SLC3A1* gene was identified via whole-genome sequencing (WES) and Sanger sequencing on the proband. Bioinformatic analysis was carried out to predict the pathogenic function. Findings from our study have clinical implications on the genetic screening of cystinuria and expand the mutation spectrum of *SLC3A1*, which would contribute to the progression of gene therapy for cystine stones.

## 2. Materials and Methods

### 2.1. Proband and Clinical Treatment

A three-generation family containing 7 family members was recruited in this study. Our study was approved by the local Ethics Committee of The Children’s Hospital, Zhejiang University, School of Medicine (2021-IRBAL-068). A 5-year-old Chinese girl was recruited from The Children’s Hospital of Zhejiang University School of Medicine, Hangzhou, China, in April 2021. She was admitted to hospital with complaints of abdominal pain for 4 months and was diagnosed with urolithiasis. X-ray radiography and computed tomography (CT) were used to detect the size of the calculi. An ultrasound-guided percutaneous nephrolithotomy (PCNL) was performed to remove the stones.

### 2.2. Genetic Analysis

WES of the proband was performed using DNA extracted from peripheral blood cells. Genomic DNA was extracted from peripheral EDTA-treated blood and amplified by using the CWE2100 Blood DNA Kit (Cwbio, Beijing, China). The resulting amplicons were prepared into libraries by using XGen Hybridization and Wash Kit (IDT, USA) and KAPA Hyper Prep Kit (KAPA biosystems). The libraries were sequenced on a NextSeq CN500 sequencer (GloriousMed Co. Ltd., Shanghai, China).

The selected candidate variant was validated by using the Sanger sequencing technique on an ABI-3730xL DNA Analyzer (Life Technologies, USA). The specific PCR primers (forward (c.818G>A) 5′-AAGGATCAGGGAGGGCAATGA-3′, reverse (c.818G>A) 5′-TGTCTGAGAGT CTACATTTGCACCA-3′ and forward (c.1011G>A) 5′-GGAGGTGTGGGAGTCGCTAAAT-3′, reverse (c.1011G>A) 5′-AGGGTACTTGCTAGACACAGGAC-3′), and Phanta^®^ Max Super-Fidelity DNA Polymerase (Vazyme, Nanjing, China) were used for the amplification of target gene. The amplicon was verified by 2% agarose gel electrophoresis.

### 2.3. Bioinformatic Analysis of Variants

The mutated gene detected from the patient’s family is *SLC3A1* (NM_000341.4), which encodes the rBAT protein. The structure of the heterodimer of rBAT and b^0, +^AT in open confirmation was downloaded from the Protein Data Bank (PDB ID: 6LI9). The Pymol visualization tool (http://www.pymol.org/, accessed on 20 July 2022) was used to delineate the structure of SLC3A1. The multiple sequence alignments were carried out by using Consurf server and Unipro UGENE. Combined Annotation Dependent Depletion (CADD) (https://cadd.gs.washington.edu/snv, accessed on 28 July 2022) is a predictive software with an integration algorithm, and the C-scores were calculated based on the analysis of deleteriousness of single-nucleotide substitution to perform a precise assessment. Rare exome variant ensemble learner (REVEL) (https://sites.google.com/site/revelgenomics/, accessed on 28 July 2022) is an ensemble method that is used to predict pathogenic mutations based on SIFT, PROVEAN, PolyPhen, and Mutation Taster.

## 3. Results

### 3.1. Clinical Information

A Chinese family with three generations, including seven people, was described in this research (Figure 1A). The index patient is a five-year-old child who was diagnosed with urolithiasis. Urine biochemical examinations of the proband and her parents indicated that the proband has a high level of cystine and renal damage index, whilst her parents are at normal (Table 1). The radiography and computed tomography (CT) scan showed an area of 19.8 mm × 19.6 mm calculus in the left pelvic kidney (Figure 2A,B,D). During the first surgery, the proband’s left kidney was placed with an 18 cm 4.8 Fr double-J stent retrogradely. Multiple amber-colored stones with hard textures were found in the calyx and pelvis (Figure 3A–C). Using 400 μm Holmium laser fiber, the stones were fragmented at a setting of 1.8 J and 12 Hz. Postoperative stone analysis revealed that the stones were mainly composed of cystine (Figure 3D). Postoperative CT scan showed dense punctate shadows in the left renal pelvis and calyx, the largest of which was approximately 9.1 mm × 11 mm (Figure 2C,E). In May 2021, the patient underwent a secondary operation, in which the patient underwent a left percutaneous nephrostomy with no calculi observed. After double-J stent replacement, two calculi in the left middle and lower calyx were observed by using ureteroscopy, the sizes of which are 8 mm × 5 mm and 3 mm × 2 mm, respectively. The stones were fragmented by subsequent performance of holmium laser lithotripsy. After surgery, the double-J stent and catheter were left in place and no calculus was observed (Figure 2F). The double-J stent was removed 2 weeks after discharge. It is noted that the patient has a family history of cystine stones, for which the proband’s father (II-1) was diagnosed with cystine urolithiasis at the age of 25.

### 3.2. Genetic Findings

After performing WES analysis, we found that the index patient carried two heterozygous pathogenic variants (c.818G>A and c.1011G>A) within the *SLC3A1* gene (Figure 1A). The variant c.818G>A results in the change of 273 amino acids from cysteine to tyrosine. This novel variant was neither documented by the Genome Aggregation Database (gnomAD), nor reported by previous studies. The variant c.1011G>A results in the synonymous mutation of proline. This variant has been reported in five heterozygous individuals in gnomAD. For variants validation, Sanger sequencing was carried out among other family members and revealed that c.1011G>A in exon 5 of the gene *SLC3A1* was detected in grandmother (I-4) and mother (II-2); c.828G>A in exon 4 of the gene *SLC3A1* was detected in grandfather (I-1) and father (II-1) (Figure 1B).

### 3.3. Bioinformatic Analysis

The rBAT protein encoded by *SLC3A1* possesses three domains: a cytoplasmic topological domain (1-87 aa), a transmembrane domain (TM, 88-108 aa), and an extracellular topological domain (109-685 aa) (Figure 4A) [19,20]. The mutation p. Cys273Tyr is located in exon 4 and disrupts β-strand (272-275 aa) and breaks a disulfide bridge with cystine at site 242 [20]; while the mutation p. Pro337Pro is located at the end of exon 5 (Figure 4A) [21]. The mutation of p. Cys273Tyr was found to be near the end of domain B (Figure 4A). The structure of SLC3A1 was displayed in the cartoon and the mutation site (p. Cys273Tyr) was represented in the magenta sphere (Figure 4B).

Sequence conservative analysis revealed that the sequences of rBAT exhibited high conservation among various species (Figure 5A). The cystine^273^ showed as a buried residue, while the proline^337^ was an exposed residue (Figure 5B). To note, the cystine^273^ is in the middle of a predicted functional residue and a structural residue (Figure 5B).

Combined annotation-dependent depletion (CADD) and rare exome variant ensemble learner (REVEL) values were calculated to evaluate the pathogenicity of these mutations, and the REVEL score was used to evaluate the missense variant exclusively. The higher score of CADD (>20) and REVEL (0.5) reflect greater likelihood of causing disease. The CADD values of the mutation c.818G>A and c.1011G>A is 23.9 and 27.3, respectively (Table 2). The REVEL value of the mutation c.818G>A is 0.764 (Table 2), indicating these two mutations are deleterious.

## 4. Discussion

Cystine stone is a genetic disease with a high recurrence rate and no curative treatment [22,23]. Currently, the treatment of cystine stones in clinical events is symptomatic relief rather than a fundamental elimination of stone generation. Apart from surgery, there are three main therapeutic strategies for cystine stones: fluid intake increase, urine alkalization, and cystine binding [24]. High fluid intake and urine pH are capable of increasing the solubility of cystine. Increasing fluid intake to maintain urinary cystine at a concentration below 250 mg/L enables insoluble cystine crystals to dissolve in urine [11]. Potassium citrate is an optimal drug which is used to alkalize urine at a pH of 7.5–8.0, given that it has no effect on sodium output [11,25]. Cystine is composed of two cysteines which were connected by a disulfide bond. Thiol drugs such as tiopronin and D-penicillamine are sulfhydryls which can cleave cystine into two cysteines via binding to disulfide bonds and form complexes with cysteines. The solubility of these complexes is 50 times than that of cystine. Therefore, accumulated cystine can be excreted from urine [26].

Heterodimeric amino acids transporters (HATs) are expressed at the apical membrane of the small intestine and proximal renal tubules [19]. They are responsible for cystine and dibasic amino acid reabsorption and exchanging neutral amino acids simultaneously [19]. HATs are composed of two subunits, rBAT and b^0, +^AT, which are encoded by *SLC3A1* and *SLC7A9,* respectively. rBAT is responsible for trafficking and stability, whereas b^0, +^AT appears to be more important in substrate binding [20,27]. Mutations in either subunit would result in dysregulated cystine reabsorption, which leads to the accumulation of cystine in urine [28].

In this study, a proband from a three-generation Chinese family was described. By using the WES approach, a double heterozygous variant in the gene *SLC3A1* (NM_000341.4: c. 818G>A, p. Cys273Tyr; c. 1011 G>A, p. Pro337Pro) was identified, of which c. 818.G>A was a novel variant. A 9-year-old male patient who suffered from nine operations to remove a cystine stone was reported to have a heterozygous mutation (c.817T>C and c.1097A>G) located at the Cys273 and Gln366 site of *SLC3A1,* respectively [17]. Two cystinuria cases with c. 1011G>A from Quebec and France have been reported previously [29,30]. Notably, no stone formation was detected in these two cases. In vitro study exhibited that this synonymous variant caused a skipping of exon 5 splicing [30]. By using MaxEntScan technique, this abnormal splicing was predicted to alter the strength of the reference 5′ splicing site [30].

The amino acid sites of 273 and 337 in SLC3A1 are highly conserved among all species, thereby implying that mutations may cause severe pathological changes (Figure 5A). Additionally, the cystine^273^ is in the middle of a predicted functional residue and a structural residue (Figure 5B). To evaluate the pathogenicity of the missense discovered in this case, the REVEL and CADD score has been calculated (Table 2). The CADD score (>20) and REVEL score (>0.5) indicate both mutations are pathogenic.

In previous studies, the clinical presentation of the patients with the mutation of c.1011G>A, p.Pro337Pro is only cystinuria, while the patient with the mutation of c.817T>C, p. Cys273Arg exhibited stone formation (17, 29, 30). Moreover, the synonymous mutation c.1011G>A, p. Pro337Pro does not cause any structural change. Hence, we hypothesized that the mutation of c.818G>A, p. Cys273Tyr and c.1011G>A, and p.Pro337Pro work in coordination with each other to contribute to recurrent cystine stone generation in this case.

Different kinds of *SLC3A1* mutations are recorded in the HGMD database. Missense/nonsense takes up 61%; gross deletions take up 14.3%; small deletions take up 9.0%. The rest of the mutations are small indels (1.0%), small insertions (5.2%), splicing (6.2%), regulatory (0.5%), gross insertions/duplications (2.4%), and complex rearrangements (0.5%) (Figure 6). In this study, the two genetic mutations found in the proband were missense/nonsense variants. Since missense/nonsense takes up the majority of *SLC3A1* mutation, genetic screening of cystine stone patients is an urgent need for clinical management.

However, a limitation in our study is that we lack wet-lab experiments to explore the functional change of the mutation of interest. In vivo models are required to detect the function of mutated proteins such as ion transport, acid-base balance, crystallization, and anti-inflammation. Induced pluripotent stem cells (iPSC) can also be generated from patients to mimic the pathogenic microenvironment for confirmation in a pathogenic circumstance. The combination of computational approach and wet-lab validation enables us to obtain a deep insight into hereditary diseases.

## 5. Conclusions

Our study identified a novel heterozygous mutation of *SLC3A1* for cystinuria based on WES screening of a 5-year-old case and indicated that a gene-mutation-based screening strategy should be considered in the clinic to provide a promising approach for cystine stone diagnosis and clinical management.

## Figures and Tables

**Figure 1 genes-13-02173-f001:**
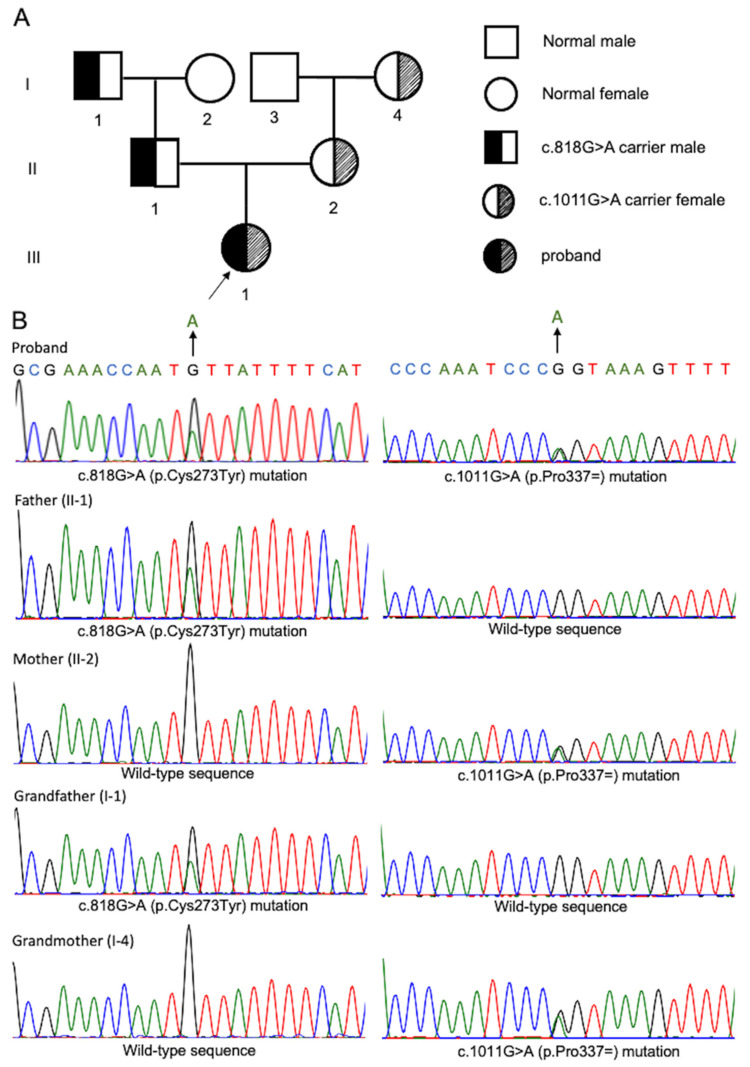
Pedigree of the cystine stone patient’s family and Sanger sequencing validation. (**A**) A pedigree diagram of three generations of this family. Half-filled symbols indicate carriers with c.818G>A mutation; half-shadowed symbols indicate carriers with c.1011G>A mutation. Black arrow indicates the proband. (**B**) The identified mutations that were validated via Sanger sequencing. In all the panels, the locations of mutant bases are indicated by arrows.

**Figure 2 genes-13-02173-f002:**
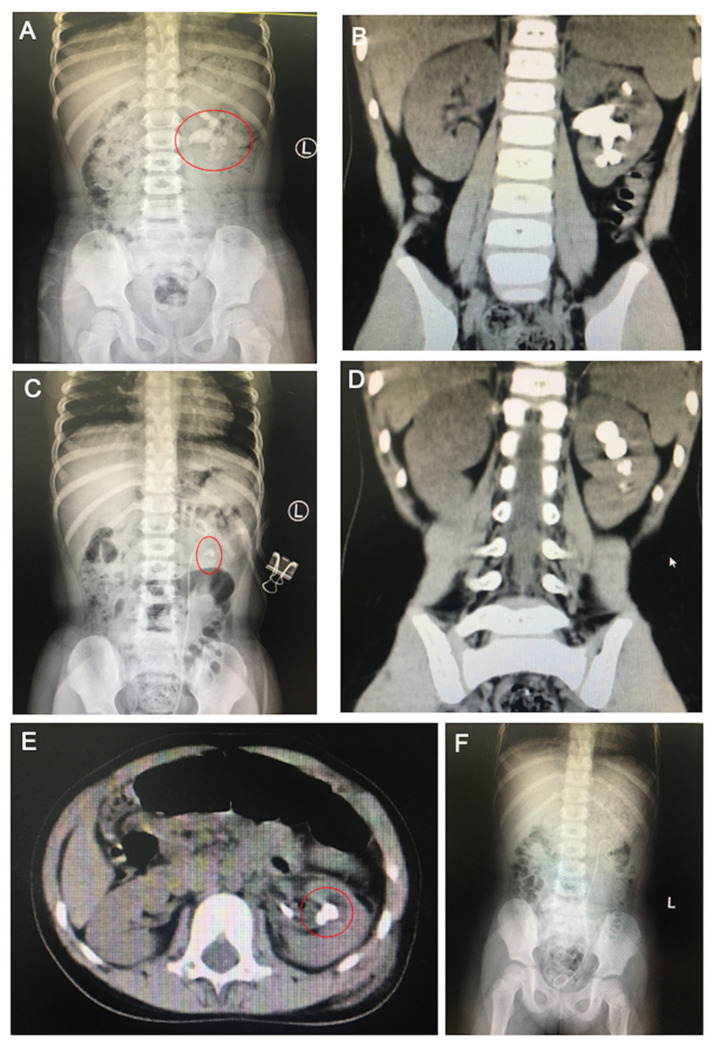
The imaging tests of the patient (**A**,**B**,**D**) Preoperative abdominal radiography and computed tomography (CT) revealed multiple calculi in the left kidney. (**C**,**E**) Postoperative abdominal radiography and CT scan showed a retained stent and two calculi after one-stage operation in the left kidney. (**F**) Abdominal radiography revealed a retained stent and no calculus in the left kidney after secondary operation.

**Figure 3 genes-13-02173-f003:**
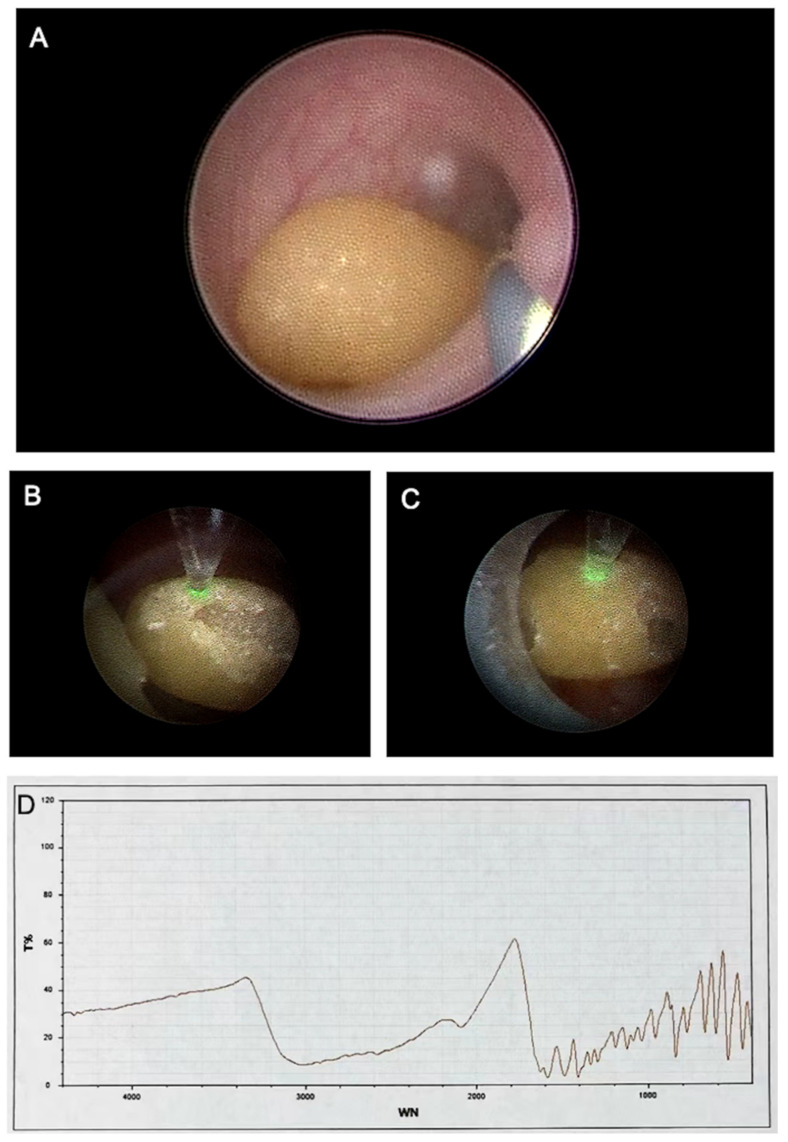
Stone removal process and component analysis. (**A**) Complete calculi visualized using a ureteroscope. (**B**,**C**) Stone targeting using a laser (green dot). (**D**) Infrared spectroscopy analysis of the stone. T: Transmittance; WN: wavelength (nanometers).

**Figure 4 genes-13-02173-f004:**
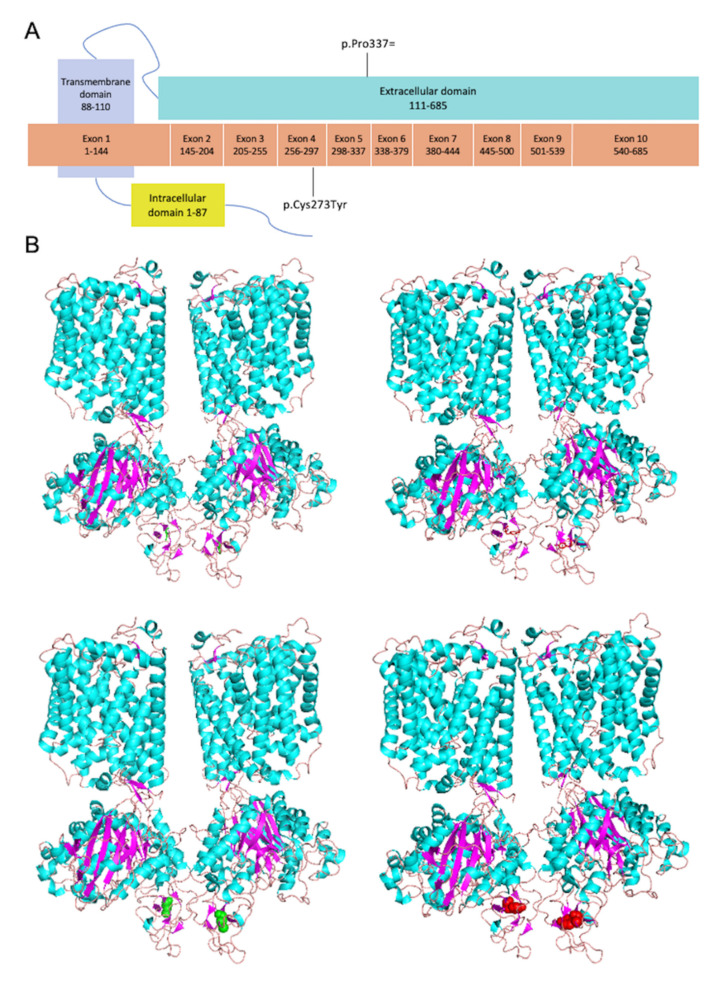
The domain and 3D structure of SLC3A1. (**A**) Distribution of proband’s mutations detected in SLC3A1 (rBAT) throughout the exons and protein domains. The rBAT protein contains 3 domains: a cytoplasmic topological domain (1-87 aa), a transmembrane domain (TM, 88-110 aa), and an extracellular topological domain (111-685 aa). The mutation p. Cys273Tyr is located in exon 4; while the mutation p.337ProPro is located at the end of exon 5. (**B**) The 3D structure of rBAT wild type (WT) (left), and rBAT p. Cys273Tyr mutation (right). The WT site was represented as a green stick/sphere (upper/lower). The mutation site was represented as a magenta stick/sphere (upper/lower).

**Figure 5 genes-13-02173-f005:**
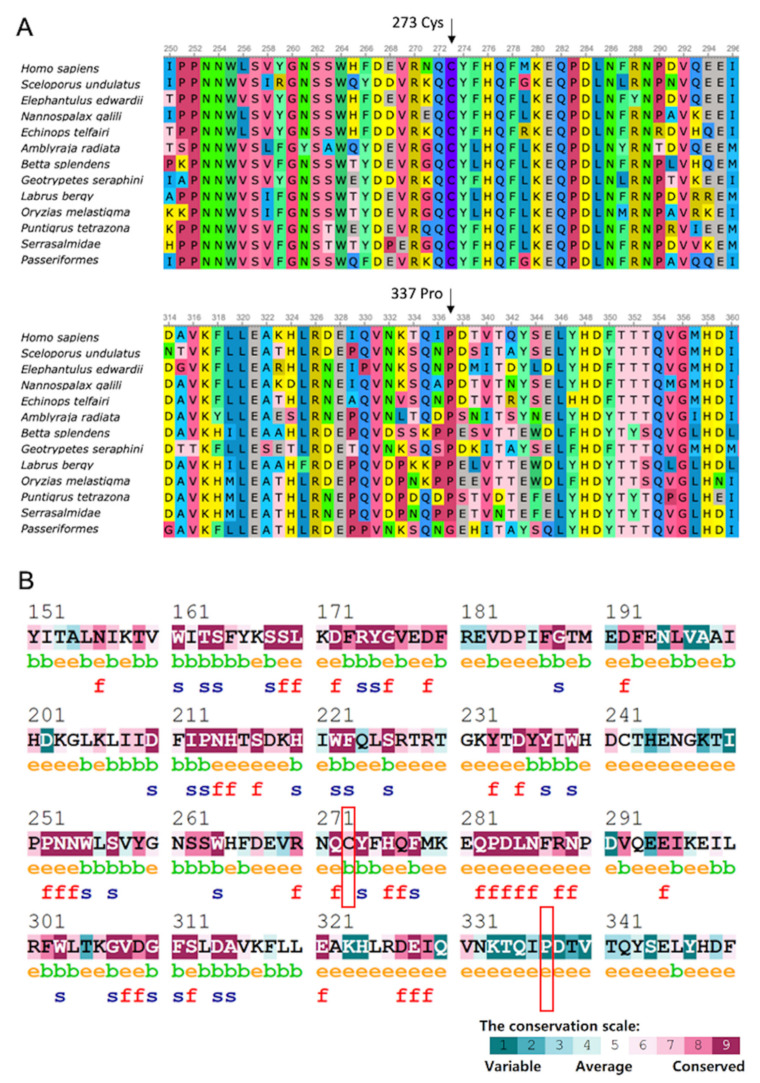
Conservation analysis of amino acid sequencing. (**A**) The mutation and neighboring sites were conserved among species. (**B**) Site 273 is highly conserved with buried residue, while 337 is a variable site with exposed residue. Colors of the ConSurf output indicate the level of sequence conservation. Purple indicates conservation and blue indicates variability. According to the neutral-network algorithm, ‘e’ indicates exposed residue; ‘b’ indicates buried residue; ‘f’ indicates predicted functional residue (highly conserved and exposed); ‘s’ indicates predicted structural residue (highly conserved and buried).

**Figure 6 genes-13-02173-f006:**
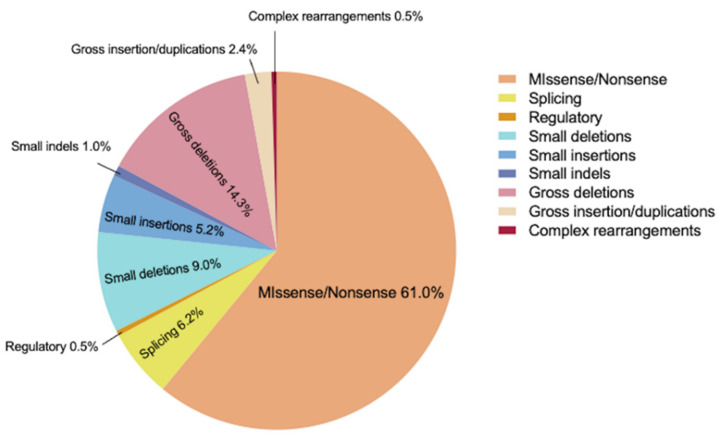
The proportion of different mutations of *SLC3A1*.

**Table 1 genes-13-02173-t001:** The urine biochemical indexes of the proband’s family. A high level of cystine and creatinine/weight was detected in proband’s urine.

	Proband (III-1)	Father (II-1)	Mother (II-2)
24-h urine volume	0.7	0.9	1
Citrate (mg/d)	819.83	239.76	466.53
Oxalate (mg/d)	45	33.22	40.81
Cystine (mg/d)	45.89↑	1.94	15.34
Creatinine/weight	49.69↑	6.81	16
Uric acid (μmol/d)	741.42	2115.23	1993.25
Creatinine (mg/d)	1341.69	545.04	1087.7
Sodium (mmol/d)	122.04	169.29	199.2
Potassium (mmol/d)	26.19	47.57	59.89
Magnesium (mmol/d)	1.35	0.52	0.58
Nitrogen (g/d)	14.34	14.48	27.72
Phosphorus (mmol/d)	10.21	14.6	12.16
Calcium (mmol/d)	0.09	0.78	0.78
pH		6.75	7.5

The red arrows indicate the proband’s cystine and creatinine/weight value are higher than standard.

**Table 2 genes-13-02173-t002:** The pathogenicity of mutated genes.

Gene	Chr	Position	Nucleotide Change	Amino Acid Change	CADD	REVEL
*SLC3A1*	Chr2	44513223	c.818G>A	p. Cys273Tyr	23.9	0.764
*SLC3A1*	Chr2	44527229	c.1011G>A	p. Pro337Pro	27.3	NA

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
