# Peer review of "Clinical Characteristics and In Silico Analysis of Cystinuria Caused by a Novel SLC3A1 Mutation"

_genes, 2022, doi:10.3390/genes13112173_

Round 1
Reviewer 1 Report
Dear Authors,
I have a suggestion on how to report the potential pathogenicity of the mutations detected in the manuscript. In the abstract (line 33) only the Cys273Tyr mutation was reported as pathogenic. However, in the results and discussion, it is stated that both mutations are deleterious based on the CADD and REVEL results (lines198 and 245). Moreover, the REVEL value is only available for C.818G> A. Thus, the reporting of deleteriousness should be corrected.
Additionally, the suggestion that appeared in the discussion that p.Cys 273Tyr is mainly responsible for recurrent cystine stone generation does not seem appropriate. As you mentioned in the Introduction (line 65), cystinuria is an autosomal recessive hereditary disorder. So, in order for the disease to manifest itself, both alleles of the SLC3A1 gene have to be damaged. As you mentioned (line 238), c.1011G> a was proved to be connected with abnormal splicing. Therefore, it cannot be indicated that it is c.818G> A that is mainly responsible for the observed clinical symptoms.
Author Response
Dear Editors and Reviewers:
Thank you for your efforts to review our manuscript entitled “Clinical Characteristics and In Silico Analysis of Cystinuria caused by a novel SLC3A1 mutation” (ID: genes-1966699)”. We highly appreciate your time and comments that help a lot to improve the quality of the manuscript.
We have addressed the comments from reviewers in a point-by-point way. Please see them in the response letter as below.
Hopefully, the revised manuscript will meet with the criteria of publication of your well-reputed journal. Thank you in advance.
- In the abstract (line 33) only the Cys273Tyr mutation was reported as pathogenic. However, in the results and discussion, it is stated that both mutations are deleterious based on the CADD and REVEL results (lines198 and 245). Moreover, the REVEL value is only available for C.818G> A. Thus, the reporting of deleteriousness should be corrected.
Response: Many thanks for this comment. We have modified the statement in the abstract section. (Line 33-34)
- Additionally, the suggestion that appeared in the discussion that p.Cys 273Tyr is mainly responsible for recurrent cystine stone generation does not seem appropriate. As you mentioned in the Introduction (line 65), cystinuria is an autosomal recessive hereditary disorder. So, in order for the disease to manifest itself, both alleles of the SLC3A1 gene have to be damaged. As you mentioned (line 238), c.1011G> a was proved to be connected with abnormal splicing. Therefore, it cannot be indicated that it is c.818G> A that is mainly responsible for the observed clinical symptoms.
Response: Thank you for this constructive comment. We have rephrased the sentence into “Hence, we speculated that the heterozygous mutation of c.818G>A (p. Cys273Tyr) and c.1011G>A (p. Pro337Pro) collectively contribute to recurrent cystine stone generation.” (Line 254-255). In addition, the statement of the case from reference 17 was rephrased. The heterozygous mutation was added to make it more specific. (Line 236-238)
Yours sincerely,
Lexin Liu, Zihao Xu, Yuelin Guan, Ying Zhang, Xue Li, Yunqing Ren, Lidan Hu and Xiang Yan.

Reviewer 2 Report
The manuscript reports, through the description of a clinical case, a correct analysis approach aimed to perform a genetic diagnosis. In particular, the authors presented the genotypic and phenotypic characteristic of a case affected by Cystinuria, in a Chinese family composed by three generations. In my opinion the value of this study is due firstly for the meticulous characterization of the phenotype and secondly for the use of the progressive sequencing technologies integrated with a complete bioinformatics analysis.
I have only few minor comments for the authors:
Q1) In the method section, it is possible to report the gene transcript identification code of the reference sequence? I think that this would help the reproducibility of the study.
Q2) In the discussion, the authors should mention the limit of the study. The pathogenic role of the c.818G>A (p.Cys273Tyr) mutation has been inferred by the authors through a computational approach at different levels (genomic and protein levels), however it should be remembered that, in a future perspective, functional studies would be necessary to confirm the pathogenicity of the mutation.
Author Response
Dear Editors and Reviewers:
Thank you for your efforts to review our manuscript entitled “Clinical Characteristics and In Silico Analysis of Cystinuria caused by a novel SLC3A1 mutation” (ID: genes-1966699)”. We highly appreciate your time and comments that help a lot to improve the quality of the manuscript.
We have addressed the comments from reviewers in a point-by-point way. Please see them in the response letter as below.
Hopefully, the revised manuscript will meet with the criteria of publication of your well-reputed journal. Thank you in advance.
- In the method section, it is possible to report the gene transcript identification code of the reference sequence. I think that this would help the reproducibility of the study.
Response: Thank you for your suggestion. We have added the gene transcript identification code of the reference sequence to help the reproducibility of the study. (Line 114-115)
- In the discussion, the authors should mention the limit of the study. The pathogenic role of the c.818G>A (p.Cys273Tyr) mutation has been inferred by the authors through a computational approach at different levels (genomic and protein levels), however it should be remembered that, in a future perspective, functional studies would be necessary to confirm the pathogenicity of the mutation.
Response: We really appreciate your constructive comments. The addition of limitation section enriches our manuscript a lot. A paragraph to state the limitations of our study “However, the limitation of our study is that we lack wet-lab experiment to validate the functional change of the mutations. A vivo model is required to detect the function of mutated protein such as ion transport, acid-base balance, crystallization, and anti-inflammation. Moreover, the patient-derived induced pluripotent stem cells (iPSC) may be also generated in order for the confirmation in a pathogenic circumstance appropriately. The combination of computational approach and wet-lab validation enable us to get a deep insight into hereditary diseases.” was added to the discussion section. (Line 264-278)
Yours sincerely,
Lexin Liu, Zihao Xu, Yuelin Guan, Ying Zhang, Xue Li, Yunqing Ren, Lidan Hu and Xiang Yan.
